# L-Ferritin: One Gene, Five Diseases; from Hereditary Hyperferritinemia to Hypoferritinemia—Report of New Cases

**DOI:** 10.3390/ph12010017

**Published:** 2019-01-23

**Authors:** Beatriz Cadenas, Josep Fita-Torró, Mar Bermúdez-Cortés, Inés Hernandez-Rodriguez, José Luis Fuster, María Esther Llinares, Ana María Galera, Julia Lee Romero, Santiago Pérez-Montero, Cristian Tornador, Mayka Sanchez

**Affiliations:** 1Whole Genix SL., 08021 Barcelona, Spain; Beatriz.cadenas@wholegenix.com (B.C.); cristian.tornador@wholegenix.com (C.T.); 2Iron Metabolism: Regulation and Diseases Group, Josep Carreras Leukemia Research Institute (IJC), Campus Can Ruti, Badalona, 08916 Barcelona, Spain; 3Experimental Sciences and Technology Department, Universitat de Vic-Universitat Central de Catalunya, 08500 Vic, Spain; 4BloodGenetics SL, Esplugues de Llobregat, 08950 Barcelona, Spain; josepfita@hotmail.com (J.F.-T.); sperez@bloodgenetics.com (S.P.-M.); 5Pediatric OncoHematology Service, Clinic University Hospital Virgen de la Arrixaca, Instituto Murciano de Investigación Biosanitaria (IMIB), 30120 Murcia, Spain; mariam.bermudez2@carm.es (M.B.-C.); josel.fuster@carm.es (J.L.F.); mariae.llinares@carm.es (M.E.L.); anam.galera@carm.es (A.M.G.); 6Hematology Service, University Hospital Germans Trias i Pujol (HGTiP), Institut Català d’Oncologia (ICO), Badalona, 08916 Barcelona, Spain; agnesrh@iconcologia.net; 7Biomedical Engineering Department, University of Texas at Austin, Austin, TX 78712, USA; julialromero@utexas.edu; 8Program of Predictive and Personalised Medicine of Cancer (PMPPC), Institut d’Investigació Germans Trias i Pujol (IGTP), Campus Can Ruti, Badalona, 08916 Barcelona, Spain; 9Iron Metabolism: Regulation and Diseases Group, Faculty of Medicine and Health Sciences, Universitat Internacional de Catalunya (UIC), 08195 Barcelona, Spain

**Keywords:** ferritin, hereditary hyperferritinemia, hereditary hypoferritinemia, iron metabolism, cataracts syndrome, neurodegenerative disease

## Abstract

Ferritin is a multimeric protein composed of light (L-ferritin) and heavy (H-ferritin) subunits that binds and stores iron inside the cell. A variety of mutations have been reported in the L-ferritin subunit gene (*FTL* gene) that cause the following five diseases: (1) hereditary hyperferritinemia with cataract syndrome (HHCS), (2) neuroferritinopathy, a subtype of neurodegeneration with brain iron accumulation (NBIA), (3) benign hyperferritinemia, (4) L-ferritin deficiency with autosomal dominant inheritance, and (5) L-ferritin deficiency with autosomal recessive inheritance. Defects in the *FTL* gene lead to abnormally high levels of serum ferritin (hyperferritinemia) in HHCS and benign hyperferritinemia, while low levels (hypoferritinemia) are present in neuroferritinopathy and in autosomal dominant and recessive L-ferritin deficiency. Iron disturbances as well as neuromuscular and cognitive deficits are present in some, but not all, of these diseases. Here, we identified two novel *FTL* variants that cause dominant L-ferritin deficiency and HHCS (c.375+2T > A and 36_42delCAACAGT, respectively), and one previously reported variant (Met1Val) that causes dominant L-ferritin deficiency. Globally, genetic changes in the *FTL* gene are responsible for multiple phenotypes and an accurate diagnosis is useful for appropriate treatment. To help in this goal, we included a diagnostic algorithm for the detection of diseases caused by defects in *FTL* gene.

## 1. Introduction

Ferritin is an iron-binding protein that stores and releases iron and thus contributes to maintaining and controlling iron homeostasis. Iron is stored in ferritin in the Fe^+3^ form and released in the Fe^+2^ form. Tissue ferritin is a multimeric protein formed from the assembly of 24 peptide subunits, known as the light (L) and heavy (H) ferritin subunits. The H subunit has ferroxidase activity and converts iron from Fe^+2^ to Fe^+3^, which enables iron storage; it also serves to regulate pH and increase concentrations of free radicals in the body, which can be extremely damaging to cellular structures and proteins [1]. The L-ferritin subunit helps with electron transport in and out of the ferritin core protein and plays a role in iron release, as Fe^+2^, from ferritin. Unlike tissue ferritin, serum ferritin is partially glycosylated and nearly completely iron-free [2,3] and is mainly composed of L-ferritin subunits [4,5,6]. 

Focusing on the L-ferritin (*FTL*) gene, five diseases have been identified as directly resulting from mutations in this gene. These diseases include hereditary hyperferritinemia with cataract syndrome (HHCS), neuroferritinopathy, benign hyperferritinemia (or hyperferritinemia without iron overload), autosomal dominant L-ferritin deficiency and autosomal recessive L-ferritin deficiency. 

HHCS (OMIM#600886, ORPHA163) is associated with mutations located in the iron responsive element (IRE) at the 5′ untranslated region (UTR) of the L-ferritin mRNA, which results in the disruption of binding with iron regulatory proteins (IRP1 and 2), this is known as the IRP-IRE post-transcriptional regulatory system [7,8,9]. Mutations in this RNA motif result in the loss of ferritin translation repression and excess ferritin production, even though iron levels remain normal. Ferritin overproduction leads to deposits in the lens of the eye, resulting in the development of cataracts [10]. Up to now, there are 47 known mutations associated with HHCS [11]. 

In 2001, Curtis and collaborators [12] described for the first time neuroferritinopathy (OMIM#606159, ORPHA:157846), an autosomal dominant condition characterized by normal to low serum ferritin levels, progressive chorea or dystonia, and subtle cognitive deficits. Neuroferritinopathy is classified as a member of the group of diseases known as neurodegeneration with brain iron accumulation (NBIA). So far, there have been ten reported mutations causing this condition, mostly located at the C-terminal region of the *FLT* gene [13]. *FTL* mutations diminish ferritin’s ability to store iron so, in an attempt to control free iron levels, neurons produce more ferritin, resulting in iron and ferritin accumulation in the basal ganglia of the brain and leading to movement and cognitive disabilities [14]. 

Benign hyperferritinemia or genetic hyperferritinemia without iron overload (OMIM#600886, ORPHA:254704) is another *FTL* mutated disorder where patients have high (greater than 90%) glycosylated serum ferritin levels. There are three known mutations in the N-terminal region of the *FTL* gene that cause benign hyperferritinemia [15]. Despite serum ferritin hyperglycosylation, no other harmful effects have been detected in patients with this disorder [16].

Finally, two variants in *FTL* have been reported causing L-ferritin deficiency, i.e., hypoferritinemia (OMIM#615604, ORPHA:440731). Mutation Glu104Ter was described in a single patient with inheritance in autosomal recessive mode and consists of a G > C nucleotide substitution in exon 3 (c.310G > T). This mutation causes a complete lack of translation of the *FTL* gene with subsequently undetectable levels of serum ferritin. This patient presented with seizures and restless leg syndrome [17]. The FTL mutation Met1Val, resulting in a change at the start codon (c.-1A > G), has also been described in a single case; however, this mutation was inherited in an autosomal dominant manner. The patient presented with decreased levels of serum ferritin, but no history of iron deficiency anemia or neurologic dysfunction [18]. 

In this study, we report the identification of two novel mutations in the *FTL* gene detected by gene sequencing. One mutation is associated with a diagnosis of HHCS and the other with a diagnosis of dominant L-ferritin deficiency. We also describe an additional dominant L-ferritin deficiency case with a previously described (Met1Val) mutation in the *FTL* gene. Moreover, we have performed an extensive review of all reported variants in the *FTL* gene linked with the previously described five conditions to help in the understanding of the phenotypes.

## 2. Results

We completely sequenced the entire coding region, intron-exon boundaries and 5′ and 3′ regulatory regions for the *FTL* gene either by Sanger sequencing or by next generation sequencing (NGS).

### 2.1. Case Studies

#### 2.1.1. Family 1—A Case with Autosomal Dominant L-Ferritin Deficiency

Proband II.1 from family 1 (Figure 1A and Table 1) is a four-year-old female of Spanish origin referred to the department of Pediatric OncoHematology Service of the Clinic University Hospital Virgen de la Arrixaca because of refractory hypoferritinemia (serum ferritin 4–9 ng/mL) unresponsive to oral iron supplementation, without any accompanying sign or symptoms. Physical examination was normal with normal weight and size for age. At the age of six, the proband complained of recurrent severe headaches. Cerebral CT and MRI were normal except for a small (non-specific) subcortical area of gliosis in the right frontal lobe. This was considered an incidental finding by pediatric neurologists who established a provisional diagnosis of primary headache and recommended treatment with flunarizine.

Mutation analysis revealed the presence of T > A change in the intron 3 of the *FTL* gene, at position 375 + 2 (NM_000146.3:c.[375 + 2T > A];[=], HGSV nomenclature). This variant was also found in a heterozygous state in the father of the proband (I.1). Human Splicing Finder software predicted that this variant alters the wild type splicing donor site, affecting mRNA splicing. This mutation is novel and has not been previously reported either in the literature or in any public database (ENSEMBL, NCBI, 1000Genomes, public HGMD). However, another variant without a known clinical significance exists at the same position, c.375 + 2T > C, reported in the SNP database as rs1371561306 and with a very low allele frequency (MAF = 0.000008 reported in TOPMED database).

#### 2.1.2. Family 2—A Case with Autosomal Dominant L-Ferritin Deficiency

Proband II.1 from family 2 (Figure 1B and Table 1) is an asymptomatic two-year-old girl evaluated at the department of Pediatric OncoHematology Service of the Clinic University Hospital Virgen de la Arrixaca for further investigation of mild neutropenia and eosinophilia. The patient had been previously diagnosed with a small ventricular septal defect. During her follow-up, we did not find neutropenia or other hematological anomalies, but rather marked hypoferritinemia without anemia. Hypoferritinemia was unresponsive to oral iron supplementation. She occasionally complained of mild asthenia and an occasional mild headache, which were found to be tension headaches after a full evaluation by a pediatric neurologist. The proband´s mother had low serum ferritin levels (<6 ng/mL), low transferrin saturation (9.6%) with normal levels of transferrin.

Sequencing analysis of this proband (Figure 1B, II.1) showed an A > G substitution at position 1 in the heterozygous state, causing the start codon methionine to change into valine. This mutation was previously reported in 2004 [18] and was described in the SNP database as rs139732572 with a very low allele frequency (MAF = 0.000008 reported in the ExAc database). This variant has been classified in ClinVar database as pathogenic (Variation ID 96689), causing L-ferritin deficiency in dominant inheritance mode. Here we report the second case of a patient with hypoferritinemia and this same mutation in the *FTL* gene (NM_000146.3:c[1A > G];[=], NP_000137.2:p(Met1Val);(=), HGSV nomenclature). 

#### 2.1.3. Family 3—A Case with HHCS

The proband I.1 in family 3 (Figure 1C and Table 1) is a 65-year-old man with a history of enolism and dyslipidemia, showing high levels of serum ferritin (>3000 ng/mL), motive for what he was referred to the Hematology Service at the University Hospital Germans Trias i Pujol (HGTiP). At age of 45, he underwent cataract surgery. Initially, he underwent three therapeutic phlebotomies, but they were suspended due to the development of anemia. Magnetic resonance (MR) imaging showed normal deposits of liver iron (30 μmol/g). The family history suggests the presence of HHCS due to the existence of additional cases of hyperferritinemia (proband´s son and uncle) and cataracts (proband´s mother). The proband’s son (II.1) is a 39-year-old male with history of stage 1 orchiectomized and disease-free seminoma and no surgical removed cataracts. He was contacted by the same hematology service (HGTiP) under suspicion of HHCS because of hyperferritinemia (>2000 ng/mL) and cataracts. The hematological evaluation was normal except for the high ferritin levels, and liver magnetic resonance showed normal levels of hepatic iron (20 µmol/g). 

The genetic studies performed on family 3 showed the presence a heterozygous deletion c.[-164_-158del7] located in the 5′ *FTL* IRE in the proband (I.1) and his son (II.1), both affected with hereditary hyperferritinemia with cataracts syndrome (Figure 1). Genetic analyses were not available for the mother and the uncle of the proband. This variant consists of a deletion of seven nucleotides (CAACAGT), excising part of the hexanucleotide loop and upper stem of the *FTL* IRE (Figure 2). Following the traditional nomenclature for *FTL* IRE mutations, we refer to these mutations as Esplugues +36_42del7 mutation (HGVS nomenclature as NM_000146.3:c.[-164_158del7];[=]). This deletion is predicted to impair the IRE structure. RNA secondary structure modelling of WT and mutated *FTL* 5′ IRE sequences was performed using the Sfold web server, which predicted that -164_-158del7 mutation, located at the hexanucleotide loop, is likely to disturb the WT IRE conformation (Figure 3). In addition, the SIREs web server prediction [19] indicate loss of the IRE structure, as the mutated query returned no results. This mutation has not been previously described in the literature, but other similar IRE deletions have been previously demonstrated to be pathogenic for HHCS [11]. The location and severity of this mutation, together with the clinical manifestations of HHCS present in the affected individuals of this family, indicates that this variant is most probably the genetic cause of disease in this family.

### 2.2. Update on L-Ferritin Mutations and Diseases

Mutations in L-ferritin gene causes five different phenotypic diseases, as summarized in Table 2. After an extensive literature search, we have collected all 63 different mutations reported so far for these five diseases, including the two novel mutations reported in the present work (Figure 2 and Appendix A).

Most of the *FTL* mutations described correspond to hereditary hyperferritinemia with cataract syndrome (HHCS), including 36 point mutations, 9 deletions, and 2 insertion-deletions. All the mutations for HHCS are located at the 5´UTR of the *FTL* gene (chr19:49468566-49468764), affecting the primary sequence and structure of *FTL*-IRE (Figure 2). HHCS is an autosomal dominant disorder, and all reported variants are in heterozygous state except for three cases [11,21,22] where homozygous mutations have been described. Patients with HHCS do not present clinical manifestations other than high serum ferritin levels and congenial bilateral nuclear cataracts.

Neuroferritinopathy is the only NBIA disorder with an autosomal dominant inheritance. It is caused by mutation in the *FTL* gene. Up to now, nine insertions from one to 16 nucleotides located in the exon 4 of *FTL* gene have been reported to cause this neurodegenerative disorder (Figure 2). These insertions alter the reading frame of the C-terminal region, generating a longer protein with additional aberrant amino acids. In addition, a missense Ala96Thr mutation (rs104894685) has been also described to cause neuroferritinopathy; amino acid 96 is predicted to be situated at the same tertiary structure region as the pathogenic insertions [13]. Mutations in the C-terminal region of FTL disrupt α-helix D or E, which seems to be essential for the stability of the peptide [23,24]. Clinical and biochemical manifestations of this disease include low serum ferritin levels, iron accumulation in the basal ganglia and progressive and severe neurological dysfunctions with subtle cognitive deficits in some cases.

Three heterozygous mutations in *FTL* exon 1 have been associated with hyperferritinemia without iron overload where cataracts were absent; this condition has also been named benign hyperferritinemia. The Thr30Ile mutation (rs397514540, ExAc MAF = 0.000008) has been identified in French and British families [15,25]. Two further pathogenic mutations—p.(Ala27-Val) and p.(Gln26Ile)—have been reported in two additional patients [26]. Mutations in exon 1 of *FTL* are associated with higher than normal serum ferritin glycosylation. These three variants alter the A α-helix near the N terminus of L-ferritin; it has been hypothesized that the aberrant peptide extends the length of the hydrophobic cluster of amino acids at the N terminus, increasing the secretion of L-ferritin [15]. However, the reason for the development of hyperferritinemia and hyperglycosylation associated with these mutant ferritin forms is still not fully elucidated [25].

We have described in this report a novel intronic splicing mutation in the *FTL* gene (NM_000146.3:c.[375 + 2T > A];[=]) associated also with autosomal dominant L-ferritin deficiency. Including this novel splicing mutation, two heterozygous mutations have been described to cause L-ferritin deficiency with autosomal dominant transmission. Cremonesi and collaborators identified in 2004 a heterozygous A > G substitution in the first nucleotide of FTL, which change the ATG start codon (methionine) into a valine [18]. This mutation is predicted to encode a non-functional and unstable protein. Despite the low serum levels of L-ferritin, the proband presenting this mutation does not show either serious neurological problems (other than headaches) indicating that the molecular mechanism of L-ferritin deficiency with autosomal dominant inheritance is haploinsufficiency. 

Finally, only one Italian case has been reported to cause L-ferritin deficiency with autosomal recessive inheritance; a homozygous substitution at nucleotide 310 G to T that produces a premature stop codon (E104X) [17]. This amino acid change is predicted to be located in the middle of the α-helix C domain, a critical region for the stability of the protein. In silico analysis predicted that a stop codon at this position produces a truncated protein unable to fold into a functional peptide and, therefore, leads to the generation an L-ferritin subunit with a complete loss of function. This homozygous mutation is associated with a more aggressive phenotype, which is characterized by undetectable ferritin levels, idiopathic generalized seizures and atypical restless leg syndrome.

## 3. Discussion

L-ferritin disease is a clear case of Mendelian disease-related genes that are associated with multiple diseases, including hyperferritinemia linked with cataracts in HHCS, isolated hyperferritinemia (benign hyperferritinemia), hypoferritinemia associated with other symptoms (neurological symptoms as in neuroferritinopathy, or muscular symptoms in autosomal recessive L-ferritin deficiency), or isolated hypoferritinemia (autosomal dominant L-ferritin deficiency). For improving the diagnosis of these different diseases, we have created a clinical diagnostic algorithm of diseases caused by mutations in *FTL* gene (Figure 4).

The underlying mechanisms for these disease patterns are not fully elucidated for all these diseases. In HHCS, it is known that the pathological molecular mechanism is linked to the disruption of the IRP-IRE post-transcriptional regulatory system, with the de-repression of L-ferritin mRNA and the subsequent overproduction of L-ferritin protein that precipitates and deposits in the lens of the eyes, producing cataracts. In HHCS, authors have argued that an association exists between the clinical severity of the disease and the location of the mutation in *FTL*-IRE [11,27,28]. Higher serum ferritin levels are mostly associated with mutations in the apical loop or the C-bulge area, compared with mutations in the upper or lower stems. These findings are consistent with our cases in family 3, i.e., a father and son with a seven-nucleotide deletion in the IRE of the *FTL* gene (5′-CAACAGU3′). The deletion partially affects the hexanucleotide loop of the IRE, and patients show considerably high serum ferritin levels (2071–3037 ng/mL) and develop early bilateral cataracts (Table 1).

Mutations in exon 4 of *FTL* gene that alter the C-terminal sequence and the length of the protein have been reported to cause neuroferritinopathy. These mutations affect the D- or E α-helix (see Figure 2), leading to significant disruption of the tertiary and quaternary structure of the FTL protein and producing an unstable and leaky ferritin. Studies have shown that mutant ferritin maintains the normal spherical shell structure and size [29]. However, mutant FTL C-termini, rich in amino acids with iron-binding properties, may be extended and unravel out of the shell. These mutant structures could initiate aggregation, forming ferritin inclusion bodies/precipitates [29,30]. Previous in vitro functional analysis and in vivo mouse models have revealed that these molecular-level defects have two main consequences: (1) the loss of normal protein function, reducing iron incorporation; (2) and the acquisition of toxic function through radical production, ferritin aggregation, and reactive oxygen species (ROS) generation [14,31,32,33]. It has been demonstrated that mutations in the C terminus have a dominant-negative effect, which explains the dominant transmission of the disorder [34]. 

In benign hyperferritinemia, missense variations in exon 1 of *FTL* increase the hydrophobicity of the ferritin N-terminal site. The A-α-helix of the protein is altered and hyperglycosylated, although the precise molecular mechanism is not clear yet. 

L-ferritin deficiency is characterized by haploinsufficiency of the FTL protein in autosomal dominant L-ferritin deficiency, or complete loss of function in autosomal recessive L-ferritin deficiency. Mutations associated with this condition are predicted to affect protein expression. The inactivation of one allele of *FTL*, which occurs in the two described heterozygous cases of autosomal dominant L-ferritin deficiency, produces a significant reduction of L-ferritin in serum. The phenotype of this dominant condition is normal apart from low levels of serum ferritin and low transferrin saturation. However, the homozygous mutation Glu104Ter causes a total loss of function of L-ferritin and leads to ferritin missing the FTL chain. Previous mutational studies have suggested that the presence of H homopolymer ferritin in fibroblasts is associated with reduced cellular iron availability and increased ROS production, which triggers cellular damage. These findings were also found in neurons derived from patient fibroblasts and correlate with the neurological phenotype of this more severe condition [17]. 

In this study, we have identified two new cases of autosomal dominant L-ferritin deficiency, one due to a new mutation in intron 3 of the *FTL* gene that most probably affects splicing. The T > G substitution in nucleotide 375 + 2 (genomic coordinates g.49469665 in Assembly GRCh37.p13) modifies a highly conserved dinucleotide GT donor site, a key sequence recognized by the spliceosome during splicing. Mutations in splice site junctions are likely to lead to exon skipping or total or partial intron retention in the mRNA transcript in most cases [32]. Our in silico analysis using Human Splicing Finder suggests that our variant would activate a new alternative donor site onwards (375 + 5G and +6T), leading to the insertion of four additional nucleotides in the mRNA sequence between exon 3 and 4 and altering the reading frame of the transcript [33]. Therefore, this change would lead to a premature stop codon formation and the putative generation of a truncated FTL protein of 128 amino acids, if the aberrant mRNA is not detected and degraded by the nonsense mediated decay (NMD) system. According to our bioinformatics prediction, the c.375 + 2T > A mutation will not generate a dominant-negative version of FTL protein, but a truncated FTL protein completely missing the E- α-helix domain and partially lacking the D- α-helix domain. Supported by the clinical manifestations found in this family (low serum ferritin levels, low transferrin saturation and lack of serious neurological or movement abnormalities), the molecular mechanism in this case is most probably due to the loss of function of FTL and it will be not expected that the disease derives in neuroferritinopathy. However, we cannot totally and completely exclude the later development in life of brain iron overload and neuroferritinopathy in these patients by a yet unknown and novel mechanism. 

Patients with hereditary hyperferritinemia could be misdiagnosed as patients suffering from hereditary hemochromatosis, liver dysfunction or inflammation. Some patients have received unnecessary invasive diagnostic techniques, such as liver biopsy, and are inappropriately treated with venesections and phlebotomies that can cause severe iron-deficiency anemia. On the other hand, clinical manifestations of neuroferritinopathy including tremor, parkinsonism, psychiatric problems, and abnormal involuntary movements, the presence of ferritin-iron precipitation in glia cells and neurons [30], are often misdiagnosed and treated as Huntington’s or Parkinson’s disease. Potential treatment targets for neuroferritinopathy may include an optimized iron chelator to induce the re-solubilization of iron aggregations, in combination with radical scavengers to prevent oxidative ferritin damage [35]. Patients with benign hyperferritinemia could be misdiagnosed as patients with HHCS due to the presence of high ferritin levels and the possible late-onset appearance of cataracts. Apart from the surgical removal of cataracts in HHCS, HHCS and L-ferritin deficiency have no specific therapy.

These facts emphasize the importance of a correct and early genetic diagnosis for the subsequent implementation of proper treatment, avoiding detrimental or inappropriate treatments. Following clinical algorithms, such as the one included in this publication (Figure 4) and also HIGHFERRITIN Web Server (http://highferritin.imppc.org/) [36] will surely help in this goal. 

## 4. Materials and Methods 

### 4.1. Patients

All subjects gave their informed consent for inclusion before they participated in the study. The study was conducted in accordance with the Declaration of Helsinki, and the protocol was approved by the Ethics Committee on 10th July 2015.

### 4.2. DNA Extraction, PCR Amplification, and DNA Sequencing

Genetic studies were performed with minor differences for all the pedigrees. Genomic DNA was extracted from peripheral blood using the FlexiGene DNA kit (Qiagen) according to manufacturer’s instructions. 

Ferritin gene regions (exonic, intron–exon boundaries, and untranslated regions) were sequenced using the Sanger method or next generation sequencing (NGS).

For the Sanger method, the FTL gene was amplified using 50 ng of genomic DNA. Primer sequences and PCR conditions are available upon request. The resulting amplification products were verified on a 2% ethidium bromide agarose gel. The purified PCR products were sequenced using a conventional Sanger method. Sequencing results were analyzed using Mutation Surveyor software (SoftGenetics LLC, PA, USA).

For NGS methods, patients were analyzed using a targeted NGS gene panel (v14) for hereditary hemochromatosis and hyper/hypoferritinemia that included the following nine genes: *HFE*, *HFE2*, *HAMP*, *TFR2*, *SLC40A1*, *BMP6*, *FTL*, *FTH1*, and *GNPAT*.

Briefly, the capture of genomic regions was conducted starting from 225 ng of gDNA using a custom design HaloPlex^TM^ Target Enrichment 1–500 kb kit (Agilent Technologies, Santa Clara, CA, USA) according to the manufacturer’s instructions. Library quality was determined using the Agilent 4200 TapeStation System. The quantity was measured using a QubitdsDNA assay kit with a Qubitfluorometer (Life Technologies, Carlsbad, CA, USA) and fluorescence was detected using a SpectraMax Gemini EM microplate reader (Molecular Devices). Libraries were sequenced using MiSeq reagent kit v2 (300 cycles) (Illumina, San Diego, CA, USA) on an Illumina or a MiSeq or MiniSeq sequencer (Illumina, San Diego, CA, USA), generating 150-bp paired-end reads. Samples were aligned with the reference human genome GRCh37/hg19 and data analysis was performed using our algorithms. Mutations detected by NGS were confirmed by conventional Sanger sequencing. 

Genetic variants are reported following the official Human Genome Variation Sequence (HGVS) nomenclature and refer to NM_ 000146.3 for the *Homo sapiens FTL* transcript variant and NP_ 000137.2 for the *Homo sapiens* FTL protein.

Reported mutations in this study have been submitted to the Leiden Open Variation Database (http://www.lovd.nl) or to ClinVar (http://www.ncbi.nlm.nih.gov/clinvar).

### 4.3. FTL RNA Fold Predictions

RNA folding analysis was carried out to predict the IRE structure of wild-type (WT) and mutated *FTL*-IRE using the Sfold web server (http://sfold.wadsworth.org/) [20]. DNA sequences used for folding predictions are shown in Appendix A. The SIREs web server tool was also used to predict iron-responsive elements [19].

## Figures and Tables

**Figure 1 pharmaceuticals-12-00017-f001:**
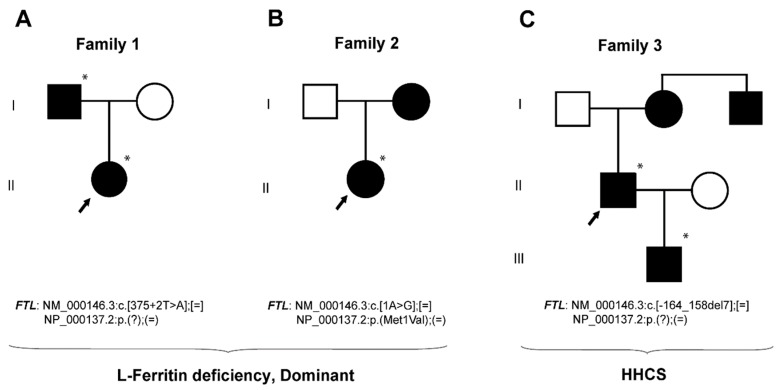
Pedigree trees from three studied families affected from dominant L-ferritin deficiency and HHCS. Squares indicate males and circles females. Probands are pointed with an arrow. Filled symbols indicate affected members and asterisks indicate subjects with genetic studies done at BloodGenetics SL. Mutations are named according the HGVS nomenclature.

**Figure 2 pharmaceuticals-12-00017-f002:**
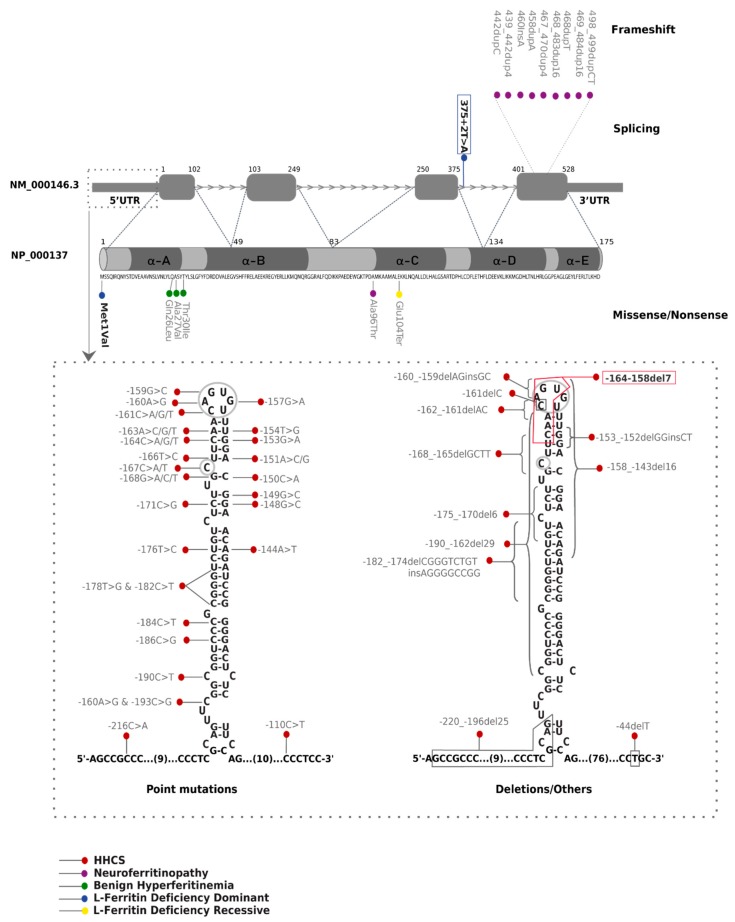
Schematic localization of literature reported and new *FTL* mutations. Mutations described in this work are in bold and new mutations are boxed. The domains of the five alpha helices (A to E) are represented in the protein (NP_000137.2). Mutations are classified as nonsense, frameshift, missense, or splicing. Here we report FTL protein changes using the three-letter amino acid code.

**Figure 3 pharmaceuticals-12-00017-f003:**
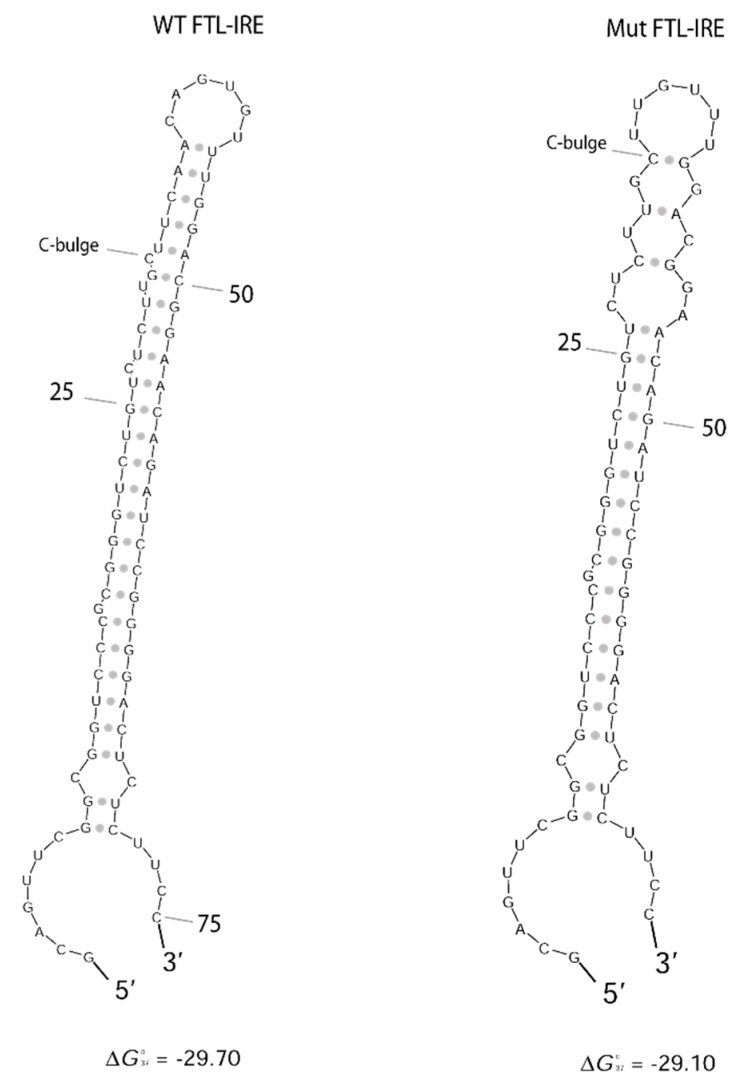
WT and mutated *FTL*-IRE fold prediction. Predicted secondary structure of WT and mutated *FTL*-IRE using Sfold web server [20]. Deletion in hexanucleotide loop (c.-164_158del7) is expected to disturb completely the IRE structure. Nucleotides are numbered from the transcription starting site. Free energy (ΔG) is detailed.

**Figure 4 pharmaceuticals-12-00017-f004:**
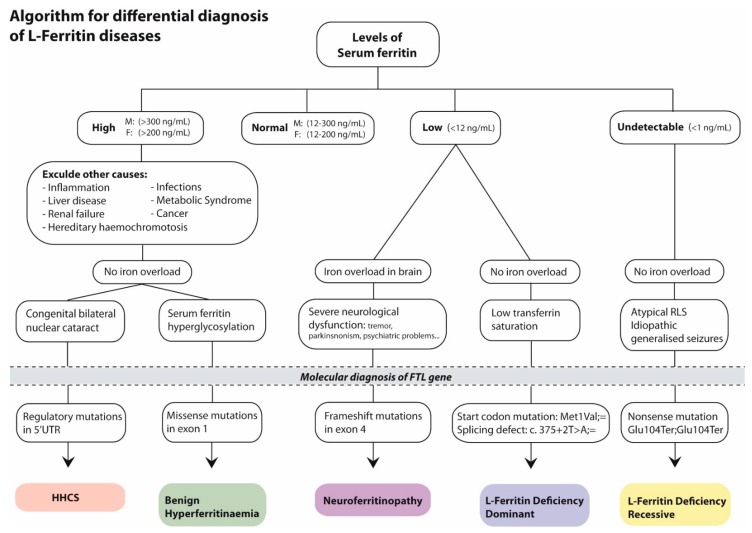
Algorithm for diagnosis of diseases caused by defects in *FTL* gene. The following abbreviations were used: F, female; M, male. The mutation nomenclature used follows the HGVS guidelines.

**Table 1 pharmaceuticals-12-00017-t001:** Genetic and clinical features of the probands.

Case	Family 1	Family 2	Family 3	Reference Values
Patient	II.1	II.1	II.1	-
Gender	F	F	M	-
Age at diagnosis (years)	4	2	67	-
Hb (g/dL)	12.2–13.3	13.1–13.7	14.0	13.5–17.5 (M). 12.1–15.1 (F)
MCV (fL)	78–84	80	90.2	80–95
Ferritin (ng/mL)	4–9	2–7	3037	12–300 (M), 12–200 (F)
Transferrin sat (%)	12.9	17.2–26.2	22.0–41.0	25–50
Iron (µL/dL)	n/a	61.95	46	49–226
Mutation	c.375 + 2T > A	p.Met1Val	c.-164_158del7	-
	Novel	Previously reported [18]	Novel	-
Disease	L-ferritin deficiency	L-ferritin deficiency	HHCS	-
Inheritance	AD	AD	AD	-

The following abbreviations were used: HB, hemoglobin; MCV, mean corpuscular volume; TF sat, transferrin saturation; F, female; M, male; n/a, not available. The mutation nomenclature used follows the HGVS guidelines.

**Table 2 pharmaceuticals-12-00017-t002:** Summary of diseases caused by mutations in *FTL* gene.

Disease	Hereditary Hyperferritinemia Cataract Syndrome	Benign Hyperferritinemia	Neurodegeneration with Brain Iron Accumulation 3	L-Ferritin Deficiency, Dominant	L-Ferritin Deficiency, Recessive
**First publication**	[7,8]	[15]	[12]	[18]	[17]
**Inheritance**	Autosomal dominant	Autosomal dominant	Autosomal dominant	Autosomal dominant	Autosomal recessive
**Mechanism**	LOST OF IRP REGULATION	(DO NOT PROCEED)	DOMINANT NEGATIVE EFFECT	HAPLOINSUFICIENCY	TOTAL LOSS OF FTL
**Mutation/s**	Many in the 5′ IRE	Missense in exon 1	Frameshift in exon 4	p.(M1V; =)	p.(E104X; E104X)
**Type**	5′ UTR	Affects the A α-helix near the N-terminus	Predicted to cause loss of the C-terminal secondary structure	Start loss	Nonsense
**Hematological features**	**High serum ferritin** Normal serum iron Normal transferrin saturation Normal red cell counts Normal hematologic parameters	**High serum ferritin** Serum ferritin hyperglycosylation Normal hematologic parameters	**Low serum ferritin**	**Low serum ferritin** Low transferrin saturation (17%) Normal serum iron Normal hematologic parameters	**Undetectable serum ferritin** Normal Transferrin saturation Normal hematologic parameters
**Other features**	Congenital bilateral nuclear cataract		Severe neurological dysfunction Gastrointestinal dysphagia		Idiopathic generalized seizures Atypical RLS Progressive hair loss

The following abbreviations were used: IRP, iron regulatory protein; IRE, iron responsive element; UTR, untranslated region; RLS, restless leg syndrome; A α-helix, first domain alpha helix of FTL protein.

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
