# Peer review of "L-Ferritin: One Gene, Five Diseases; from Hereditary Hyperferritinemia to Hypoferritinemia—Report of New Cases"

_pharmaceuticals, 2019, doi:10.3390/ph12010017_

Round 1

Reviewer 1 Report

Herein Cadenas and co-workers describe the identification and characterization of three novel mutations in the L-ferritin gene resulting in either ferritin deficiency (2 case) or hyperferretninema cataract syndrome.

This is a very well performed study which in addition provides a comprehensive and balanced  uptodate overview of L- ferritin gene mutations.

Minor: 

1)      Case 1 and 2 were identified by low ferritin levels and failing response to oral iron. Did the patients also receive intravenous iron and what was the hematological response to that? Was a defect in dietary iron absorption also excluded (oral iron absorption test, genetic analysis of specific mutations DMT!/TMPRSS6?)

2)      In cases 1 to 3 is there any information available whether or not H  ferritin mRNA and protein expression were altered in primary cells of patients (such as monocytes)?

3)      The authors nicely explain the pitfalls in the diagnosis /misdiagnosis of patietns with hypo and hyperferritinemia. It would be most helpful the reader if they could draw an algorithm of how to diagnostically approach and best identify such patients as the majority of subjects have other causes for ferritin  deficiency or hyperferritinemia.

4)      Please briefly state in the final part that there is no specific therapy for either HHCS or L-ferritin deficiency.  

Author Response

REVIEW 1:

We thank the reviewers for their comments and for their thorough revision of our manuscript. We have performed a complete revision of our manuscript ID pharmaceuticals-423331 according to their suggestions. For a better visualization and reviewer’s inspection, we have highlighted all new changes introduced in this new version.

1)      Case 1 and 2 were identified by low ferritin levels and failing response to oral iron. Did the patients also receive intravenous iron and what was the hematological response to that? Was a defect in dietary iron absorption also excluded (oral iron absorption test, genetic analysis of specific mutations DMT1/TMPRSS6?)

RESPONSE:

Patients 1 and 2 had normal hemoglobin levels (no anemia) and they did not receive intravenous iron. No specific test for dietary iron absorption evaluation were carried out (apart from excluding celiac disease through anti-transglutaminase IgA normal value).

TMPRSS6 and DMT1 genes were analyzed in the NGS panel and no pathogenic mutations were found.

2)      In cases 1 to 3 is there any information available whether or not H ferritin mRNA and protein expression were altered in primary cells of patients (such as monocytes)?

RESPONSE:

Unfortunately, we don´t have information about H ferritin mRNA and protein expression in primary cells in any reported patient of this publication.

3)      The authors nicely explain the pitfalls in the diagnosis /misdiagnosis of patients with hypo and hyperferritinemia. It would be most helpful the reader if they could draw an algorithm of how to diagnostically approach and best identify such patients as the majority of subjects have other causes for ferritin deficiency or hyperferritinemia.

RESPONSE:

Thank you for this relevant suggestion. We have now included in the manuscript an algorithm for the diagnosis of L-ferritin diseases, Figure 4, and we mention it in the Discussion section pages 9-12 lines 250-252, 337-338 and 368-370.

4)      Please briefly state in the final part that there is no specific therapy for either HHCS or L-ferritin deficiency. 

RESPONSE: We have included the following sentence in page 11 line 333:

“Apart from the surgical removal of cataracts in HHCS, HHCS and L-ferritin deficiency have no specific therapy.”

Reviewer 2 Report

Excellent review. I only have minor comments:

1) Please, could you clarify why patients with complete lack of translation of ferritin may still show some level of Ft in the serum (i.e. line 84-85)...is this FtH?

2) Figure 2: please use larger dots to clearly discriminate the color code associated with each group of disease.

3) Some periods are duplicated (line 31) or a space is missing (line 66)

4) please, could you speculate why some dominant mutations are such? For instance, the Authors indicate on line 307-311 that a non dominant-form is probably generated as a consequence of the aberrant splicing. Please, elaborate. 

Author Response

REVIEW 2:

We thank the reviewers for their comments and for their thorough revision of our manuscript. We have performed a complete revision of our manuscript ID pharmaceuticals-423331 according to their suggestions. For a better visualization and reviewer’s inspection, we have highlighted all new changes introduced in this new version.

1) Please, could you clarify why patients with complete lack of translation of ferritin may still show some level of Ft in the serum (i.e. line 84-85)...is this FtH?

RESPONSE:

For a better understanding and to address the comment of the reviewer, we have improved the writing of the mentioned sentences as it was not fully correct. In the introduction, line 85-86 we have change:

Before: “This mutation causes a complete lack of translation of the FTL gene with subsequently low serum and tissue levels of ferritin.”

Now: “This mutation causes a complete lack of translation of the FTL gene with subsequently undetectable levels of serum ferritin.”

2) Figure 2: please use larger dots to clearly discriminate the color code associated with each group of disease.

RESPONSE: Done. We have improved the design of figure 2 by a whole amplification of the imagen.

3) Some periods are duplicated (line 31) or a space is missing (line 66)

RESPONSE: Done. We have revised the punctuation in the whole paper.

4) please, could you speculate why some dominant mutations are such? For instance, the Authors indicate on line 307-311 that a non dominant-form is probably generated as a consequence of the aberrant splicing. Please, elaborate.

RESPONSE:

In neuroferritinopathy there is a dominant negative effect as the resulted FTL protein is aberrant and longer extending the Ct part. A dominant negative effect, where the defective protein interferes with the wild-type protein, is normally seen in proteins that form polymers/multimers as it is the case of ferritins forming a multimeric complex with 24 subunits of FTL and FTH proteins (in different ratios depending on the tissue).

Genetically both neurodegeneration with brain iron accumulation 3 (also named neuroferritinopathy) and Dominant L-ferritin deficiency are inherited with a dominant inheritance pattern. However, the molecular mechanism in both diseases is different as in neuroferritinopathy we have a dominant negative effect as mutations are generating mutated FTL proteins that interferes with the wild-type FTL forms, whereas Dominant L-ferritin deficiency we have a loss of function of one allele or an haploinsufficiency where the presence of one single correct allele is not sufficient and the disease appears.

We have rephrased line 311 to 318 in the manuscript (see page 10 of the new version of the manuscript) for a better understanding of the effect of the splicing mutation c.375+2T>A.
